# No One Representation to Rule Them All: Overlapping Features of Training Methods

**Raphael Gontijo-Lopes, Yann Dauphin & Ekin D. Cubuk**
Google Research, Brain Team
`{iraphael,ynd,cubuk}@google.com`

## Abstract

Despite being able to capture a range of features of the data, high accuracy models trained with supervision tend to make similar predictions. This seemingly implies that high-performing models share similar biases regardless of training methodology, which would limit ensembling benefits and render low-accuracy models as having little practical use. Against this backdrop, recent work has developed quite different training techniques, such as large-scale contrastive learning, yielding competitively high accuracy on generalization and robustness benchmarks. This motivates us to revisit the assumption that models necessarily learn similar functions. We conduct a large-scale empirical study of models across hyper-parameters, architectures, frameworks, and datasets. We find that model pairs that diverge more in training methodology display categorically different generalization behavior, producing increasingly uncorrelated errors. We show these models specialize in subdomains of the data, leading to higher ensemble performance: with just 2 models (each with ImageNet accuracy ˜76.5%), we can create ensembles with 83.4% (+7% boost). Surprisingly, we find that even significantly low-accuracy models can be used to improve high-accuracy models. Finally, we show diverging training methodology yield representations that capture overlapping (but not supersetting) feature sets which, when combined, lead to increased downstream performance.

## 1 Introduction

Over the years, the machine learning field has developed myriad techniques for training neural networks. In image classification, these include data augmentation, regularization, architectures, losses, pre-training schemes, and more. Such techniques have highlighted the ability of networks to capture diverse features of the data: textures/shapes (Geirhos et al., 2018), robust/non-robust features (Ilyas et al., 2019), and even features that fit a random, pre-determined classifier (Hoffer et al., 2018).

Despite this representation-learning power, methods that yield high generalization performance seem to produce networks with little behavior diversity: models make similar predictions, with high-accuracy models rarely making mistakes that low-accuracy models predict correctly (Mania et al., 2019). Additionally, the quality of features learned (e.g.: for downstream tasks) seems dictated by upstream performance (Kornblith et al., 2019). Finally, training on subsets of the data yields low-accuracy models that don't make performant ensembles (Nixon et al., 2020). This seemingly suggests that high-performing models share similar biases, regardless of training methodology.

Without behavior diversity, ensemble benefits are limited to reducing noise, since models make correlated errors (Perrone & Cooper, 1992; Opitz & Maclin, 1999). Without feature diversity, representations might not capture important features for downstream tasks, since feature reuse has been shown to be crucial for transfer learning (Neyshabur et al., 2020). Without knowing the effect of training methodology, one might conclude that low-accuracy models have no practical use, since their predictions would be dominated by high-accuracy ones.

One open question is if these findings faced unavoidable selection bias, since the highest-performing models have historically been trained with similar supervised objectives on IID datasets. Up until recently, this hypothesis was difficult to test. That changed with the recent success of large-scale contrastive learning, which produces competitively-high accuracy on standard generalization and robustness benchmarks (Radford et al., 2021; Jia et al., 2021). This motivates revisiting the question:

*How does training methodology affect learned representations and prediction behavior?*

To settle these questions, we conduct a systematic empirical study of 82 models, which we train or collect, across hyper-parameters, architectures, objective functions, and datasets, including the latest high performing models CLIP, ALIGN, SimCLR, BiT, ViT-G/14, and MPL. In addition to using different techniques, these new models were trained on data collected very differently, allowing us to probe the effect of both training objective, as well as pre-training data. We categorize these models based on how their training methodologies diverge from a typical, base model and show:

1. Model pairs that diverge more in training methodology (in order: reinitializations → hyper-parameters → architectures → frameworks → datasets) produce increasingly uncorrelated errors.

2. Ensemble performance increases as error correlation decreases, due to higher ensemble efficiency. The most typical ImageNet model (ResNet-50, 76.5%), and its most different counterpart (ALIGN-ZS, 75.5%) yield 83.4% accuracy when ensembled, a +7% boost.

3. Contrastively-learned models display categorically different generalization behavior, specializing in subdomains of the data, which explains the higher ensembling efficiency. We show CLIP-S specializes in antropogenic images, whereas ResNet-50 excels in nature images.

4. Surprisingly, we find that low-accuracy models can be useful if they are trained differently enough. By combining a high-accuracy model (BiT-1k, 82.9%) with *only* low-accuracy models (max individual acc. 77.4%), we can create ensembles that yield as much as 86.7%.

5. Diverging training methodology yield representations that capture overlapping (but not super-setting) feature sets which, when concatenated, lead to increased downstream performance (91.4% on Pascal VOC, using models with max individual accuracy 90.7%).

## 2 RELATED WORK

**Diversity in Ensembles**. It is widely understood that good ensembles are made of models that are both accurate and make independent errors (Perrone & Cooper, 1992; Opitz & Maclin, 1999; Wen et al., 2020). Beyond improving ensemble performance, finding diverse solutions that equally well explain the observations can help quantify model uncertainty (also known as epistemic uncertainty) – what the model does not know because training data was not appropriate (Kendall & Gal, 2017; Fort et al., 2019). Many works have explored ways of finding such solutions (Izmailov et al., 2018). Boostrapping (Freund et al., 1996) (ensembling models trained on subsets of the data) was found not to produce deep ensembles with higher accuracy than a single model trained on the entire dataset (Nixon et al., 2020). Another work has examined the effect of augmentation-induced prediction diversity on adversarial robustness (Liu et al., 2019). More relevant to us, Wenzel et al. (2020) and Zaidi et al. (2021) have explored the effect of random hyper-parameters and architectures respectively, finding best ensembles when combining diverse models, albeit still considering similar frameworks.

**Model Behavior Similarity**. These attempts were hindered as many high-performing techniques seem to produce similar prediction behavior. Mania et al. (2019) demonstrates, via "dominance probabilities", that high-accuracy models rarely make mistakes that low-accuracy models predict correctly. This indicates that, within the models studied, high-accuracy models "dominate" the predictions of low-accuracy ones. Recht et al. (2019) shows that out-of-distribution robustness seems correlated with in-distribution performance. Relatedly, Kornblith et al. (2019) shows that upstream and downstream performance are very correlated. These jointly indicate that high-accuracy models learn strictly better representations, diminishing the importance of low-accuracy solutions (even if they are diverse). Finally, Fort et al. (2019) shows that subspace-sampling methods for ensembling generate solutions that, while different in weight space, remain similar in function space, which gives rise to an insufficiently diverse set of predictions.

**Contrastive-Learning Models; Different Large-Scale Datasets**. This model behavior similarity might be explained by the fact that the training techniques that yield high performance on image classification tasks have been relatively similar, mostly relying on supervised learning on ImageNet, optionally pre-training on a dataset with similar distribution. Recently, various works have demonstrated the effectiveness of learning from large-scale data using contrastive learning (Radford et al., 2021; Jia et al., 2021). They report impressive results on out-of-distribution benchmarks, and have been shown to have higher dominance probabilities (Andreassen et al., 2021). These represent some of the first models to deviate from standard supervised training (or finetuning) on downstream data, while still yielding competitive accuracy. They add to the set of high-performing training techniques,

which include data augmentation (Cubuk et al., 2018; 2020), regularization (Srivastava et al., 2014; Szegedy et al., 2016; Ghiasi et al., 2018), architectures (Tan & Le, 2019; Dosovitskiy et al., 2020; Hu et al., 2018; Iandola et al., 2014; Li et al., 2019; Szegedy et al., 2016; Simonyan & Zisserman, 2014; Sandler et al., 2018), losses (Chen et al., 2020; Radford et al., 2021; Jia et al., 2021), pre-training schemes (Kolesnikov et al., 2020; Pham et al., 2021), and provide the motivation for revisiting the question of whether training methodology can yield different model behavior.

## 3 METHOD

### 3.1 MODEL CATEGORIZATION

In order to evaluate the performance of learned representations as a function of training methodology, we define the following categories, which classify model pairs based on their training differences:

1. **Reinits**: identical models, just different in reinitialization.
2. **Hyper-parameters**: models of the same architecture trained with different hyper parameters (e.g.: weight decay, learning rate, initialization scheme, etc).
3. **Architectures**: models with different architectures, but still trained within the same framework and dataset (e.g.: ResNet and ViT, both with ImageNet supervision).
4. **Frameworks**: models trained with different optimization objectives, but on same dataset (e.g.: ResNet and SimCLR, respectively supervised and contrastive learning on ImageNet).
5. **Datasets**: models trained on large-scale data (e.g.: CLIP or BiT – trained on WIT or JFT).

In some sense, model categories can be supersets of one another: when we change a model architecture, we may also change the hyper-parameters used to train such architecture, to make sure that they are optimal for training this new setting. Unless stated otherwise, all ensembles are comprised of a fixed base model, and another model belonging to one of the categories above. This way, each category is defined relative to the base model: model pairs in a given category will vary because Model 2 is different than the base model along that axis. The result is that as we navigate along model categories (Reinit → ... → Dataset), we will naturally be measuring the effect of *increasingly dissimilar training methodology*. See Appendix Table 1 for details.

### 3.2 MODEL SELECTION

We collect representations and predictions for 82 models, across the many categories above. We fix ResNet-50, trained with RandAugment, as our base model. ResNet is a good candidate for a base model since it is one of the most typical ImageNet classification models, and the de-facto standard baseline for this task. In total, we train/collect models in the categories: 1) Reinit; 2) Hyper-parameters (51): varying dropout, dropblock, learning rate, and weight decay, sometimes jointly; 3) Architectures (17): including EfficientNet, ViT, DenseNet, VGG; 4) Framework (2): including SimCLR, and models trained with distillation; and 5) Dataset (12): including CLIP, ALIGN, BiT, and more, trained on WIT (Radford et al., 2021), the ALIGN dataset, JFT (Sun et al., 2017), etc. We additionally collect high-performing models MPL, ALIGN (L-BFGS), ViT-G/14, BiT-1k, CLIP-L, EfficientNet-B3 for some of our analyses. These are some of the latest, highest-performing models for ImageNet classification.

We found it necessary to calibrate all models using temperature scaling (Roelofs et al., 2020; Guo et al., 2017) to maximize ensemble performance. Finally, unless stated otherwise, we only use models in the same narrow accuracy range (74-78% accuracy on ImageNet), which guarantees that the effects observed are indeed a function of diverging training methodology, and not of any given model being intrinsically more performant than another. A complete list of models can be found in the Appendix.

### 3.3 ENSEMBLING

In our paper, we use ensembling as a tool for understanding: as such, our goal is **not** to find methods to ensemble the highest accuracy models and reach state-of-the-art. Instead, we wish to use ensembling to probe whether/when training methodology yields uncorrelated (and therefore useful) predictions.

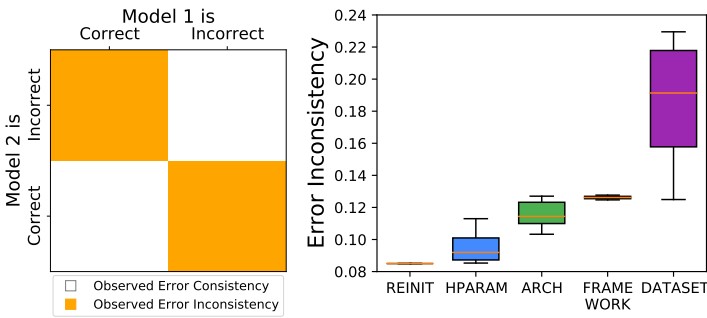

Figure 1: **As training methodology diverges** (Reinit → Dataset), **errors become uncorrelated**. *Left*: 'Observed error inconsistency' is the fraction of examples where only one model in the pair makes a correct prediction. Higher error inconsistency indicates the model errors are uncorrelated. *Right*: As models are trained with increasingly different methodologies, their error inconsistency grows, providing an opportunity for converting these examples into correct ensemble predictions.

## 4 RESULTS

In order to understand whether model similarity reported in literature varies as a function of training methodology, we evaluate error correlation by measuring the number of test-set examples where one model predicts the correct class, and the other predicts an incorrect class. We call this **Error Inconsistency**, as it complements the error consistency measure, defined in Geirhos et al. (2020).

We choose error inconsistency to measure error correlation because it allows us to connect the ensemble prediction diversity (which we are interested in quantifying) directly into performance. That is, when errors are consistent (i.e.: both models make a correct prediction, or both models make an incorrect prediction), this agreement directly translates into higher/lower accuracy after ensembling. When errors are inconsistent (i.e.: when one model is correct and another is incorrect), the ensemble prediction will be determined by the models' confidences. If a model is sufficiently confident, its prediction will "win" the ensembling procedure. If this model's prediction was correct, we say that this prediction was "converted" into a correct ensemble prediction.

### 4.1 AS TRAINING METHODOLOGY DIVERGES, ERRORS BECOME MORE UNCORRELATED

In Fig. 1 we see that, as we compare increasingly different training methodologies ("Reinit" → ... → "Dataset"), error inconsistency increases – the number of examples where *only one* model makes a correct prediction. This indicates that as training methodology diverges, model predictions become dissimilar and, as a result, errors become uncorrelated. As the framework and dataset categories can be orthogonal, we note that models with the most error inconsistency tend to modify both.

This also represents an opportunity for these uncorrelated mistakes to be converted into a correct prediction, when we ensemble the models. Such an opportunity will be beneficial if the conversion rate of these examples is higher than the decrease in number of examples where both models made correct predictions. As we will see in the following section, this indeed happens.

### 4.2 AS UNCORRELATED ERRORS INCREASE, SO DOES ENSEMBLE EFFICIENCY

In order to assess whether these increased uncorrelated errors can be beneficial, we measure how models in the various categories ensemble. That is, we ensemble our base model with models of the various categories, and measure the ensemble's accuracy, relative to its member models.

Per Fig. 2, as uncorrelated errors increase, so does ensemble performance (left). Additionally, because we restrict our models to ones within the narrow accuracy range 74-78%, we guarantee that this relative improvement translates into absolute improvement (center), and is not due to any individual ensemble member being intrinsically better than another. Relative to our base model ResNet-50 (76.5% on ImageNet), the most differently-trained model ALIGN-ZeroShot (75.5%) yields 83.4% top-1 accuracy when ensembled, a boost of nearly 7% accuracy.

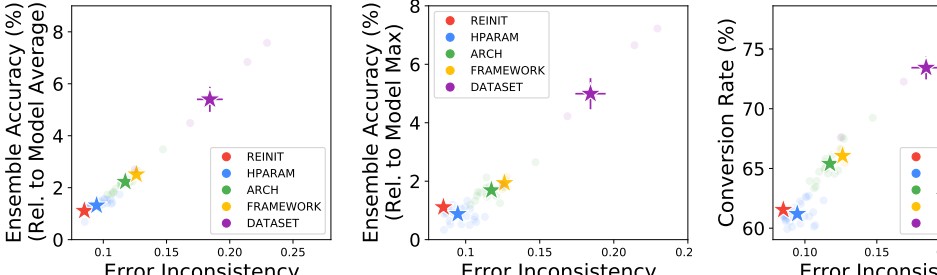

Figure 2: **As uncorrelated errors increase, so does ensemble performance** (left, center), **and so does ensemble efficiency** (right). *Left*: Error inconsistency is linearly correlated with the ensemble performance improvement, relative to the mean accuracy of the models in the ensemble. Stars represent averages over models in each category (w/ error bars). *Center*: Because we limited our analysis to models in the same 74-78% accuracy range, the increase in relative accuracy translates into absolute accuracy boost, with best performing ensembles comprising of models whose training methodologies are most different. *Right*: Surprisingly, the conversion rate – the rate at which these examples are converted into correct predictions by the ensemble – also increases. This indicates that the benefits of combining divergently-trained models go beyond increasing the number of examples that *can* become correct predictions, to also increasing how efficiently these examples *do* become correct predictions.

Taken together, these results mean that combining models with diverse training methodologies can yield performance improvements that offset the decrease in examples where both models are correct. To explain this relative boost, we also measure the conversion rate of each ensemble – the rate at which inconsistent errors are converted into correct ensemble predictions. Surprisingly, this rate also increases as training methods diverge (Fig. 2, right). This means that different training methods not only yield more opportunities for ensemble benefits, but also more efficient ensembles.

### 4.3 DISSIMILAR TRAINING METHODOLOGIES CREATE SPECIALIZED MODELS

To understand how diverging training setups create efficient ensembles, we measure the specialization of ensemble member models. We first note that, for a model's prediction to "win" the ensembling process, it only needs to be sufficiently more confident than the other model's prediction. This means that we can measure specialization by the relative confidence of two models. We do this by measuring the angle distance $\theta$ in the confidence-confidence plot (see Fig. 3, top left). When $\theta$ is high, Model 1 is more confident (relative to Model 2), and vice-versa.

In Fig. 3, we use $\theta$ to understand the specialization of different models categories. To simplify our analysis, we compare three different ensembles, from the categories Reinit (Model 2: ResNet-50), Architecture (EfficientNet-B3), and Dataset (CLIP-S), as they are representative of the spread of error correlation. When we plot a histogram of examples as a function of $\theta$, we find that: 1) As observed in Fig. 1, when training methods diverge, the models make mistakes in different sets of examples 2) As training methods diverge, Model 1 tends to be more confident in examples where only Model 1 is correct, and vice-versa for Model 2. This effect is most striking when we look at the Dataset category, where Model 2 (CLIP-S) is significantly more confident than Model 1 (ResNet-50) in examples where only CLIP-S predicts correctly, and vice-versa. This is in contrast with Reinit, where models don't seem to be significantly more confident than each other at all.

These results show: as training methodology diverges, model specialize to different data subdomains.

### 4.4 MODEL SPECIALIZATION DEPENDS ON TRAINING METHODOLOGY

Next, we want to investigate what kind of data each model category specializes to. In Fig. 4, we plot the same data as Fig. 3, but we divide the examples along their ImageNet class IDs. This allows us to inspect whether a given model is more/less specialized in a given class.

As before, we find that, while models in the Architecture category demonstrate higher specialization (Model 1 tends to be more confident in examples where it is correct, and vice versa for Model 2),

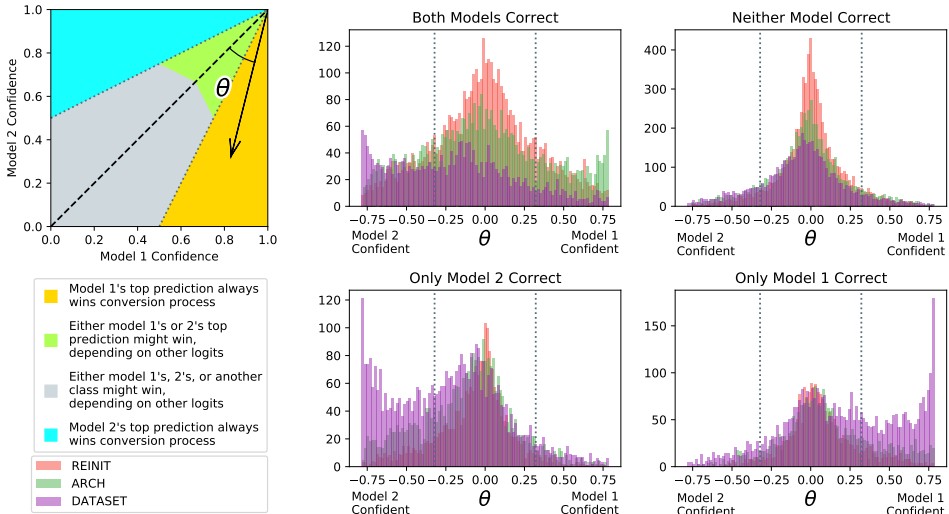

Figure 3: **Differently-trained models specialize**: We plot histograms of examples where at least one ensemble produced error inconsistency, as a function of specialization measure $\theta$, the angle distance in the confidence-confidence plot (upper left). As we saw in Fig. 1, when model training setups diverge (Reinit → Arch → Dataset), the fraction of consistent errors decreases (upper center & right), in favor of more error inconsistency (lower center & right). This added error inconsistency comes with specialization of the models in an ensemble: when only Model 1 makes a correct prediction, it is often more confident (lower right), and vice-versa for Model 2 (lower center). Faint dotted lines indicate values of $\theta$ for which a model's top-1 prediction is likely to prevail at ensemble time.

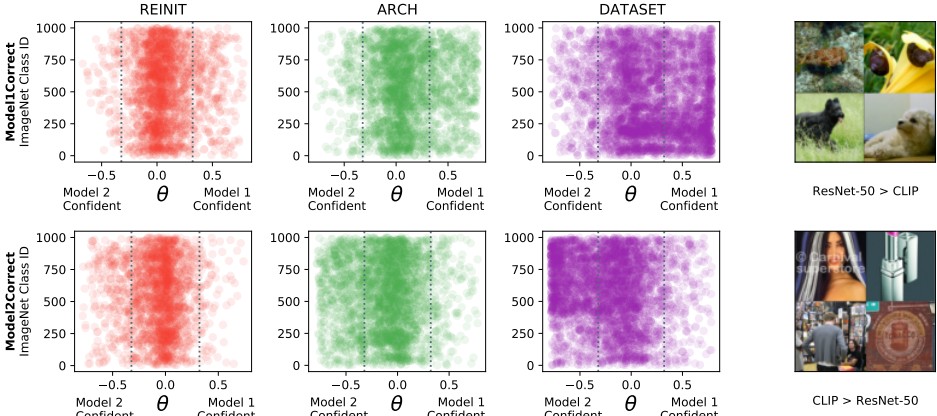

Figure 4: **Specialization type depends on training setup**: When models differ in dataset (right plot), not only is specialization highest (see Fig. 3), but also this specialization happens in different classes – CLIP is better at anthropogenic classes (cids 500-900) than ResNet-50, which is better at nature classes (cids 0-300; Right detail). When models differ in their architecture (Center plot), they are more specialized than reinitializations (Left plot), but such specialization does not correlate with specific classes.

this specialization does not seem to correlate with specific classes. In contrast, not only do models of the Dataset category display more specialization, but this specialization seems to be correlated with groups of classes. In particular, CLIP-S seems to specialize to anthropogenic images, whereas ResNet-50 seems to specialize to nature images. We suspect this phenomena is a result of CLIP-S being trained on a dataset that was collected independently, and much differently than the one used to train our base model, ResNet-50.

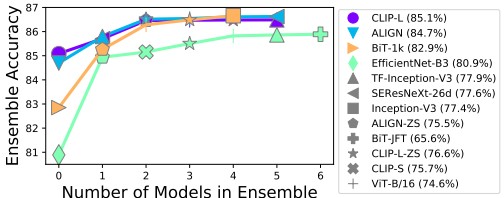

| Model 1 | Method | Model 2 | Method | Ensemble Accuracy |
|---|---|---|---|---|
| CLIP-L (85.1%) | Cont. / WIT | SEResNeXt-26d (77.6%) | Sup. / IN | 85.7% |
| ALIGN (84.7%) | Cont. / JFT | TF-Inception-V3 (77.9%) | Sup. / IN | 85.8% |
| EfficientNet-B3 (80.9%) | Sup. / IN | ALIGN-ZS (75.5%) | Cont. / JFT | 84.9% |
| BiT-1k (82.9%) | Sup. / JFT | ALIGN-ZS (75.5%) | Cont. / JFT | 83.2% |

Figure 5: **Lower-accuracy models can benefit high-accuracy ensembles**: *Left*: Starting with 4 high-accuracy models (colored markers; CLIP-L, ALIGN, BiT-1k, EfficientNet-B3), we greedily select the best lower accuracy models (each with max individual accuracy 77.9%, indicated in the legend) to ensemble, and plot the ensemble's accuracy. Colors indicate which high-accuracy model the ensemble begins with, shapes indicate which models are added. By adding only lower-accuracy models to a high-accuracy one, we are able to create ensembles that reach as high as 86.7%. This shows that lower-accuracy models can be made useful, if they are diverse enough. *Right*: In each case, the best first model to add is one that complements the base model's training methodology.

### 4.5 DIFFERENT ENOUGH LOWER-ACCURACY MODELS CAN IMPROVE ACCURACY

In classic machine learning, a common way to create high-performing ensembles is to create specialized models with bootstrapping (Freund et al., 1996), where models are trained on subsets of the data and subsequently ensembled. In deep learning, bootstrapping has not been found to work well, since training on IID subsets of the data (with similar methodology) does not seem to yield ensembles that perform better than an individual model trained on the entire dataset (Nixon et al., 2020). This seems to indicate that lower-accuracy models would have little practical benefit for performance.

In order to investigate this, and encouraged by the finding that differently-trained models can be experts in subdomains of the data, we ask whether lower-accuracy models can be beneficial. After all, some of the models in the Dataset category were trained on much larger-scale datasets (JFT, WIT), which are collected independently.

To do this, we first combine our base model, ResNet-50 (76.2%), with a lower-accuracy model trained very differently, CLIP-S-ZeroShot (63.3%, "Dataset" category), and observe a performance boost (77.7%). To push this idea further, we combine a high-accuracy model (BiT-1k, 82.85%) with *only* lower-accuracy models (max individual accuracy 77.44%), by greedily selecting ensemble members that maximize accuracy. With this procedure, we can create ensembles that yield as much as 86.66%. In Fig. 5, we repeat this procedure for 3 additional high-accuracy models and find that lower-accuracy models can consistently improve the accuracy of high-accuracy models. Importantly, the low-accuracy models need to be different enough in their training methodology (in terms of Sec. 3.1 categories): for a given high-accuracy model, the most beneficial lower-accuracy model to ensemble is one trained with a different loss and/or dataset, often both. See table in Fig. 5 for details.

This result calls into question the idea that the information learned by high accuracy models is a strict superset of those learned by low accuracy models, which we explore further in sections below.

### 4.6 SPECIALIZATION COMES FROM OVERLAPPING (BUT NOT SUPERSETTING) REPRESENTATIONS

In order to explain why differently-trained models specialize, we posit that training methodology changes the features learned by individual models. In this view, models trained differently would have an overlapping (but not supersetting) set of features which, when combined, provide more information for classification than any individual model's representation could.

We test this directly by concatenating the features of our base model (ResNet-50) with those of the models above (ResNet-50, EfficientNet-B0 and CLIP-S), and linearly evaluating these combined features on ImageNet classification. More specifically, we first randomly select features from the base model and each other model at inversely proportional rates. For example, we concatenate 25% of ResNet-50 features with 75% of CLIP-S features, yielding a final embedding that is at most the same dimensionality of the ResNet-50 features. This guarantees that any performance boost is not due to a higher number of dimensions being available, but by the quality of the features represented.

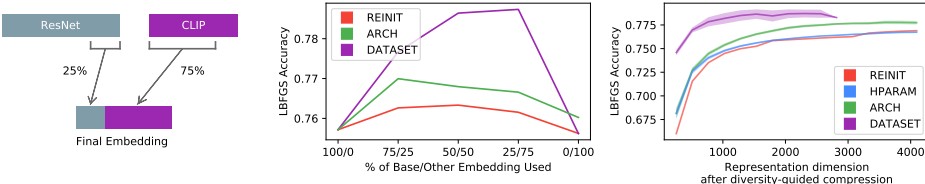

Figure 6: **Specialization comes from overlapping (not supersetting) representations**. When we combine the representations of two models at different rates (Left, 25/75 using 25% of the ResNet embedding and 75% of the other model's embedding), we find that: 1) despite being smaller number of dimensions, the concatenated embeddings (Center, e.g.: 50/50) yield higher LBFGS accuracy than the complete embedding of any of the two models (100/0 or 0/100). This indicates that each model has learned overlapping, but not supersetting, features, which when combined maximize performance. 2) when we combine embeddings of models that are more differently-trained (Center, "Reinit" $\rightarrow$ "Dataset"), we find bigger gains, indicating that divergent training induces the learning of different features. When we compress the concatenated representations using a diversity heuristic, we find that differently-trained embeddings are also more efficiently compressible, reaching higher accuracies with fewer dimensions (Right, see text for details). See Appendix for L-BFGS implementation details.

In Fig. 6 (center), we see that the best performance is obtained when features are combined. Additionally, we find that combining features yields higher performance as training methodology diverges, with the best performing combination being ResNet-50 + CLIP-S. Concurrent work similarly finds that different training objectives yield different final-layer feature embeddings (Grigg et al., 2021). This seems to confirm the idea that the methods used to train networks can generate diverse features, which capture information that neither embedding alone has captured.

To push this further, we ask how efficiently these representations capture important features of the data. To test this, we first compute the covariance of each dimension of embeddings from both models (e.g.: ResNet-50 and CLIP-S). This gives us a ranking of highest to lowest covariance dimensions, which we use as a measure of diversity. We then select features in order of most diverse to least diverse, and linearly evaluate them on ImageNet classification.

Fig. 6 (right) shows how divergently-trained models yield more diverse (and therefore more compressible) features. We tested multiples of 256 features, and found that Reinit models needed 2304 features on average to achieve 76% accuracy. In constrast, Hyperparameter models required 2057, Architecture 1392, and Dataset only required 512 features.

### 4.7 DOWNSTREAM TASK PERFORMANCE

With the knowledge that diverse training leads to diverse feature embeddings, we ask whether these can transfer better to downstream tasks. In order to maximize performance, we pick three of the highest accuracy models on ImageNet, that are also representative of very diverse training setups. Meta-Pseudo Labels (MPL) was the previous SOTA on ImageNet, reporting 90.24% top-1 accuracy (Pham et al., 2021). It is trained with a teacher network that generate pseudo labels on unlabeled data to teach a student network, whose performance on the labeled set is then used to adapt the teacher. It is trained on the JFT-300M dataset. ViT-G/14 is the current SOTA, with 90.45% top-1 accuracy (Zhai et al., 2021), when measured with EMA (Polyak & Juditsky, 1992). We obtained our image embeddings without the use of EMA, so we find effective accuracy of 88.93% It is trained supervisedly, on JFT-3B, a larger version of the JFT-300M dataset. Finally, CLIP-S is a constrastive learning model trained on the WIT dataset, yielding 75.7% top-1 accuracy (with L-BFGS linear evaluation of its features on ImageNet) (Radford et al., 2021). Despite being a lower accuracy model, as we will see, it is a useful model.

In order to test the downstream generalization ability of these models' learned representations, we linearly evaluate them on Pascal VOC (Everingham et al., 2010) using the same evaluation method as Kornblith et al. (2019). Pascal VOC is a multi-label image classification benchmark, with 20 diverse classes ranging from vehicles, animals, household objects and people. This diversity of scenes make it an interesting downstream task to study. Performance on this benchmark is measured by 11-point mAP Everingham et al. (2010).

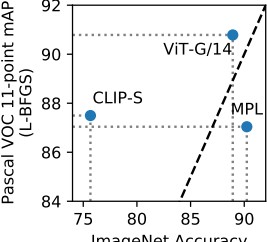 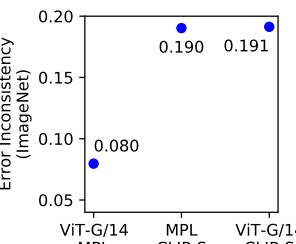 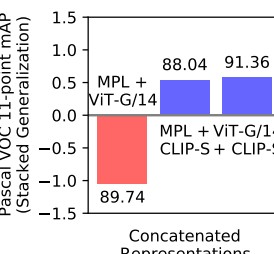

Figure 7: **Diverse training methodologies yield the best downstream performance**. *Left*: The highest-accuracy ImageNet model (MPL) is not the best-performing on Pascal VOC. *Center*: CLIP-S is the model with the most prediction diversity, among the models analyzed. *Right*: Combining models that are most diverse seems to yield the biggest boost when performing stacked generalization on a downstream task (i.e.: training a linear layer on top of 2 models' concatenated embeddings).

In Fig.7 we find that, surprisingly, the highest performing model on ImageNet (MPL) is not the best performing model on VOC. We also see that the contrastive model CLIP-S deviates from the linear trend described in Kornblith et al. (2019). This is surprising since this linear trend was previously reported with really high correlation. Indeed, when compared with the other models, CLIP-S yields the most prediction diversity on ImageNet. We additionally test how the combined features perform by concatenating them pairwise and performing linear evaluation on top of these feature combinations – known as Stacked Generalization (Wolpert, 1992). The highest performing model combinations (on VOC) are ones which combine differently trained models. Further, the best combinations do not even include MPL, which indicates that diversity can be better indicator of downstream performance than any single model's accuracy.

We posit that the reason diversely-trained model combinations yield highest performance on this downstream task is due to the diversity of features learned, which provide higher coverage of features captured that explain the data. This allows for better feature reuse, which is crucial for transfer learning (Neyshabur et al., 2020).

## 5 CONCLUSION

We have shown that diverse training methodologies, in particular, training with diverse optimization objectives on different large-scale datasets, can produce models that generate uncorrelated errors. These models ensemble more efficiently, attaining higher ensembling accuracy, since their different training setups allows them to specialize to different subdomains of the data. Due to this specialization, different-enough models can be useful for achieving high accuracies, even if they display low accuracies individually. Finally, we have also shown that they learn overlapping (but not supersetting) features, and that combining their embeddings can boost downstream performance.

The importance of behavior diversity has been highlighted in different fields. In signal processing, sensor fusion – the combination of signals from multiple sensors into a more accurate estimation – relies on these sensors making independent errors. In deep reinforcement learning, it has been hypothesized that maximizing the coverage of possible behaviors of an agent may help it acquire the skills that are useful (Eysenbach et al., 2018). In image classification, the model is our agent, its predictions the behavior, and ensembling our sensor fusion.

Our work demonstrates that this diversity of features is possible, and highlights a key question: *why didn't SGD learn it?* Perhaps there exists an objective that can produce a single best/supersetting representation but, as we've shown, none of our existing training methodologies have found it.

If we look closely, our results may also provide clues for why this is the case. To find models with categorically different generalization behavior, we had to train them with objectives that do not directly maximize the classification accuracy we are ultimately interested in. In our collective search for novelty, we have stumbled upon the decade-old finding that some problems are best solved by methods that ignore the objective, since "the objective function does not necessarily reward the stepping stones in the search space that ultimately lead to the objective" (Lehman and Stanley, 2008).

Together, our results provide encouragement for machine learning practitioners to continue creating methodologies to train high-performing models, including new frameworks & datasets, in order to expand the types of features our models learn, behaviors they exhibit, and novel ways to train them.

ACKNOWLEDGMENTS

We would like to thank the following people for their early feedback, thoughtful discussions, and help over the course of this project: Becca Roelofs, Ben Poole, Simon Kornblith, Niki Parmar, Ben Caine, Keren Gu, Ethan Dyer, Anders Andreassen, Katherine Heller, Chiyuan Zhang, Jon Shlens, Tyler Scott, Gamaleldin Elsayed, Rosanne Liu, Dan Hendrycks, Samy Bengio, Hieu Pham, Thomas Unterthiner, Chao Jia, Mostafa Dehghani, Neil Houlsby, Lucas Beyer, Josip Djolonga, Rodolphe Jenatton, Benham Neyshabur, Alec Radford, and Sam Altman.

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

## A APPENDIX

### A.1 CONCURRENT WORK

Many concurrent and recent papers have found related conclusions to ours. Here, we highlight a few of them. Abnar et al. (2021) found that when training on the same objective, as we increase the upstream accuracy, the performance of downstream tasks saturates. This is in line with our results in Sec. 4.7, that optimizing for a single objective might not capture diverse enough features for downstream performance. D'Amour et al. (2020) show that, as a result of underspecification in training, models are treated as equivalent based on their training domain performance, but can behave very differently in deployment domains.

Additionally, other recent work also finds that optimizing for diversity is beneficial: Roy et al. (2022) found that using a diverse ensemble (mixing losses and datasets) improves OOD performance in dermatology tasks. Sinha et al. (2020) explicitly optimizes for diversity of predictions using an adversarial loss, leading to improved OOD performance.

### A.2 MODEL CATEGORIZATION

In some sense, model categories are supersets of one another: when we change a model architecture, we also change the hyper-parameters used to train such architecture, to make sure that they are optimal for training this new setting. In a similar fashion, when we change the training framework (supervised to contrastive), not only do we change hyper parameters, but the architecture also changes to best suit the new learning framework. Finally, when we change dataset scales (e.g.: ImageNet →  WIT), we use the framework, architecture, and hyper parameters that allow the best performance possible on the new dataset.

### A.3 MODEL LIST PER CATEGORY

In Table 1, we list all models we used in our main analysis, Figs. 1, 2, 3, 4, 6, along with their training methodologies, calibration temperatures, individual ImageNet accuracy, error inconsistency (relative to our base model ResNet-50), and ensemble accuracy (with our base model ResNet-50). We selected these models by controlling for their individual accuracy, which helps guarantee our analysis concerns training methodology, not any individual model being inherently better.

In table 2, we list the high-accuracy models used in Figs. 5 and 7. These models are higher accuracy, and provide a stronger base for ensembling.

Table 3 lists low accuracy models, but which are trained with very different methodologies (relative to the typical model). They are therefore still useful (as we show), and were used in Sec. 4.5 and Fig. 5.

Finally, Table 4 lists other models that we trained and analyzed, but which did not reach our target accuracy range 74-78%, so were not included in the main paper analysis.

### A.4 ADDITIONAL ERROR ANALYSIS

In Fig. 8, we reproduce Fig. 1 (left), and additionally show how the number of examples where both models predict correct (center), as well as examples where both models predict incorrectly (right), both decrease. The added prediction diversity that comes with ensembling models trained differently will only be beneficial if the ensemble can efficiently convert the examples on the left, to compensate for the decreased number of examples in the center.

### A.5 L-BFGS DETAILS

In order to train the linear classifiers, we use L-BFGS in the same setup/implementation described in Kornblith et al. (2019). We train the linear heads on top of the pre-logit representations without augmentation. We find it useful to normalize each representation before any operations (concatenation, subsampling, etc). In the first part of Section 4.6, the portions of each representation are picked randomly (e.g.: random 25% dimensions of ResNet, and random 75% dimensions of CLIP), and then

| Model | Citation | Method | Prediction Head Type | Calibration Temp | ImageNet Acc | Error Inconsistency (w/ ResNet-50) | Category | Ensemble Acc (w/ ResNet-50) |
|---|---|---|---|---|---|---|---|---|
| ResNet-50 | He et al. (2016) | Sup. / IN | Trained | 1.10 | 76.20% | 0.0850 | REINIT | 77.31% |
| ResNet-101x0.5 | | Sup. / IN | Trained | 1.00 | 74.27% | 0.0972 | HPARAM | 76.73% |
| ResNet-50 (Dropout 0.1) | Srivastava et al. (2014) | Sup. / IN | Trained | 1.10 | 76.90% | 0.0853 | HPARAM | 77.23% |
| ResNet-50 (Dropout 0.2) | | Sup. / IN | Trained | 1.10 | 75.85% | 0.0883 | HPARAM | 77.27% |
| ResNet-50 (Dropout 0.3) | | Sup. / IN | Trained | 1.10 | 75.43% | 0.0910 | HPARAM | 77.12% |
| ResNet-50 (Dropout 0.4) | | Sup. / IN | Trained | 1.10 | 75.37% | 0.0928 | HPARAM | 77.12% |
| ResNet-50 (Dropout 0.5) | | Sup. / IN | Trained | 1.10 | 75.03% | 0.0965 | HPARAM | 77.06% |
| ResNet-50 (Dropout 0.6) | | Sup. / IN | Trained | 1.10 | 74.72% | 0.1006 | HPARAM | 77.01% |
| ResNet-50 (Dropout 0.7) | | Sup. / IN | Trained | 1.10 | 74.48% | 0.1058 | HPARAM | 76.89% |
| ResNet-50 (Label Smoothing 0.1) | Szegedy et al. (2016) | Sup. / IN | Trained | 0.90 | 76.22% | 0.0877 | HPARAM | 77.30% |
| ResNet-50 (Label Smoothing 0.2) | | Sup. / IN | Trained | 0.80 | 76.24% | 0.0911 | HPARAM | 77.44% |
| ResNet-50 (Label Smoothing 0.3) | | Sup. / IN | Trained | 0.80 | 75.74% | 0.0969 | HPARAM | 77.26% |
| ResNet-50 (Label Smoothing 0.4) | | Sup. / IN | Trained | 0.70 | 75.70% | 0.0996 | HPARAM | 77.31% |
| ResNet-50 (Label Smoothing 0.5) | | Sup. / IN | Trained | 0.70 | 75.51% | 0.1030 | HPARAM | 77.26% |
| ResNet-50 (Label Smoothing 0.6) | | Sup. / IN | Trained | 0.70 | 75.03% | 0.1063 | HPARAM | 77.00% |
| ResNet-50 (Label Smoothing 0.7) | | Sup. / IN | Trained | 0.70 | 74.54% | 0.1071 | HPARAM | 76.80% |
| ResNet-50 (Label Smoothing 0.8) | | Sup. / IN | Trained | 0.80 | 74.28% | 0.1130 | HPARAM | 76.98% |
| ResNet-50 (DropBlock 34, 0.9) | Ghiasi et al. (2018) | Sup. / IN | Trained | 1.00 | 74.98% | 0.0891 | HPARAM | 76.84% |
| ResNet-50 (DropBlock 1234, 0.9) | | Sup. / IN | Trained | 1.00 | 74.87% | 0.0895 | HPARAM | 76.72% |
| ResNet-50 (Learning Rate 0.05) | | Sup. / IN | Trained | 1.10 | 75.56% | 0.0871 | HPARAM | 77.10% |
| ResNet-50 (Learning Rate 0.2) | | Sup. / IN | Trained | 1.10 | 76.07% | 0.0858 | HPARAM | 77.35% |
| ResNet-50 (Weight Decay 0.00001) | Krogh & Hertz (1992) | Sup. / IN | Trained | 1.20 | 74.13% | 0.1027 | HPARAM | 76.70% |
| ResNet-50 (Weight Decay 0.00005) | | Sup. / IN | Trained | 1.10 | 75.87% | 0.0866 | HPARAM | 77.23% |
| ResNet-50 (Weight Decay 0.0002) | | Sup. / IN | Trained | 1.10 | 75.79% | 0.0865 | HPARAM | 77.16% |
| ResNet-50 (LR 0.05, WD 0.0002) | | Sup. / IN | Trained | 1.10 | 75.58% | 0.0856 | HPARAM | 77.08% |
| ResNet-50 (LR 0.2, WD 0.00005) | | Sup. / IN | Trained | 1.10 | 76.36% | 0.0864 | HPARAM | 77.49% |
| ResNet-50 (LR 1.0, WD 0.00001) | | Sup. / IN | Trained | 1.20 | 75.71% | 0.1011 | HPARAM | 77.53% |
| EfficientNet-B0 | Tan & Le (2019) | Sup. / IN | Trained | 0.90 | 76.88% | 0.1136 | ARCH | 78.65% |
| SK-ResNet-34 | Li et al. (2019) | Sup. / IN | Trained | 0.90 | 76.91% | 0.1099 | ARCH | 78.52% |
| MobileNet-V2 | Sandler et al. (2018) | Sup. / IN | Trained | 0.90 | 77.29% | 0.1033 | ARCH | 78.48% |
| VGG19 BatchNorm | Simonyan & Zisserman (2014) | Sup. / IN | Trained | 1.10 | 74.22% | 0.1232 | ARCH | 77.45% |
| Legacy-SEResNet-34 | Hu et al. (2018) | Sup. / IN | Trained | 1.10 | 74.81% | 0.1206 | ARCH | 77.70% |
| Legacy-SEResNet-50 | Hu et al. (2018) | Sup. / IN | Trained | 0.90 | 77.64% | 0.1144 | ARCH | 79.11% |
| SEResNeXt-26d | Hu et al. (2018) | Sup. / IN | Trained | 0.90 | 77.60% | 0.1187 | ARCH | 79.25% |
| DenseNet-Blur-121d | Iandola et al. (2014) | Sup. / IN | Trained | 0.90 | 76.58% | 0.1082 | ARCH | 78.22% |
| DenseNet-121 | Iandola et al. (2014) | Sup. / IN | Trained | 0.80 | 75.57% | 0.1076 | ARCH | 77.70% |
| Inception-V3 | Szegedy et al. (2016) | Sup. / IN | Trained | 1.00 | 77.44% | 0.1270 | ARCH | 79.49% |
| TF-Inception-V3 | Szegedy et al. (2016) | Sup. / IN | Trained | 1.00 | 77.86% | 0.1259 | ARCH | 79.65% |
| Adv Inception-V3 | Kurakin et al. (2018) | Sup. / IN | Trained | 1.00 | 77.58% | 0.1252 | ARCH | 79.41% |
| HRNet-W18-Small | Wang et al. (2020) | Sup. / IN | Trained | 1.00 | 75.13% | 0.1143 | ARCH | 77.78% |
| DPN-68 | Chen et al. (2017) | Sup. / IN | Trained | 1.20 | 76.31% | 0.1141 | ARCH | 78.44% |
| DLA-60 | Yu et al. (2018) | Sup. / IN | Trained | 1.10 | 77.02% | 0.1064 | ARCH | 78.35% |
| ESE-VoVNet | Lee & Park (2020) | Sup. / IN | Trained | 0.80 | 76.80% | 0.1159 | ARCH | 78.62% |
| ViT-B/16 | Dosovitskiy et al. (2020) | Sup. / IN | Trained | 1.70 | 74.55% | 0.1472 | ARCH | 78.85% |
| SimCLR | Chen et al. (2020) | Cont. / IN | Fine Tuned | 1.00 | 75.60% | 0.1277 | FWORK | 78.35% |
| ViT-DeiT-Tiny | Touvron et al. (2021) | Sup. / IN | Distilled | 1.00 | 74.50% | 0.1247 | FWORK | 77.93% |
| ViT-S/16 | Dosovitskiy et al. (2020) | Sup. / IN-21k | Trained | 0.90 | 77.86% | 0.1249 | DATASET | 79.73% |
| ALIGN-ZS | Jia et al. (2021) | Cont. / ALIGN | Zero Shot | 0.30 | 75.50% | 0.2295 | DATASET | 83.42% |
| CLIP-S | Radford et al. (2021) | Cont. / WIT | L-BFGS | 0.90 | 75.67% | 0.1687 | DATASET | 80.42% |
| CLIP-L-ZS | Radford et al. (2021) | Cont. / WIT | Zero Shot | 1.10 | 76.57% | 0.2140 | DATASET | 83.22% |

Table 1: **Models Used**. These models are all within the 74-78% accuracy range, of which we base our main results. "Sup." indicates supervised learning, "Cont." constrastive learning. "IN" indicates ImageNet.

| Model | Citation | Method | Prediction Head Type | Calibration Temp | ImageNet Acc | Error Inconsistency (w/ ResNet-50) | Category | Ensemble Acc (w/ ResNet-50) |
|---|---|---|---|---|---|---|---|---|
| EfficientNet-B3 | Tan & Le (2019) | Sup. / IN | Trained | 0.90 | 80.89% | 0.1147 | ARCH | 80.98% |
| ALIGN | Jia et al. (2021) | Constrastive / ALIGN Dataset | L-BFGS | 1.00 | 84.71% | 0.1684 | DATASET | 85.50% |
| ViT-H/14 | Dosovitskiy et al. (2020) | Sup. / JFT | Trained | 1.10 | 88.31% | 0.1646 | DATASET | 87.48% |
| ViT-G/14 | Zhai et al. (2021) | Sup. / JFT | Trained | 1.40 | 88.93% | 0.1801 | DATASET | 88.40% |
| MPL | Pham et al. (2021) | Pseudo-Label / JFT | Trained | 0.90 | 90.24% | 0.1741 | DATASET | 88.97% |
| CLIP-L | Radford et al. (2021) | Contrastive / WIT | L-BFGS | 1.00 | 85.04% | 0.1638 | DATASET | 85.36% |
| BiT-1k | Kolesnikov et al. (2020) | Sup. / JFT | Fine Tuned | 1.20 | 82.85% | 0.1593 | DATASET | 84.28% |

Table 2: **High Accuracy Models**. These models are all above the 80% accuracy. "Sup." indicates supervised learning, "Cont." constrastive learning. "IN" indicates ImageNet.

| Model | Citation | Method | Prediction Head Type | Calibration Temp | ImageNet Acc | Error Inconsistency (w/ ResNet-50) | Category | Ensemble Acc (w/ ResNet-50) |
|---|---|---|---|---|---|---|---|---|
| CLIP-S-ZS | Radford et al. (2021) | Cont. / WIT | Zero Shot | 0.01 | 63.25% | 0.2675 | DATASET | 78.01% |
| BiT-JFT | Kolesnikov et al. (2020) | Sup. / JFT | Class Mapping | 0.02 | 65.63% | 0.2744 | DATASET | 77.93% |

Table 3: **Low Accuracy Models**. These models are all below the 74% target accuracy, but they are trained with very different methodologies (relative to typical models), so we include them in a few analyses, as indicated in main text. "Sup." indicates supervised learning, "Cont." constrastive learning. "Class Mapping" indicates that the model's original JFT class vector was used, and a mapping of JFT-to-ImageNet classes was used to obtain the final prediction logits.

| Model | Method | Prediction Head Type | ImageNet Acc | Category |
|---|---|---|---|---|
| ResNet-101 | Sup. / IN | Trained | 78.42% | HPARAM |
| ResNet-50x2 | Sup. / IN | Trained | 78.70% | HPARAM |
| ResNet-50 (Dropout 0.8) | Sup. / IN | Trained | 73.68% | HPARAM |
| ResNet-50 (Dropout 0.9) | Sup. / IN | Trained | 72.48% | HPARAM |
| ResNet-50 (Label Smoothing 0.9) | Sup. / IN | Trained | 72.63% | HPARAM |
| ResNet-50 (DropBlock 34, 0.1) | Sup. / IN | Trained | 29.88% | HPARAM |
| ResNet-50 (DropBlock 34, 0.2) | Sup. / IN | Trained | 50.17% | HPARAM |
| ResNet-50 (DropBlock 34, 0.3) | Sup. / IN | Trained | 58.01% | HPARAM |
| ResNet-50 (DropBlock 34, 0.4) | Sup. / IN | Trained | 62.50% | HPARAM |
| ResNet-50 (DropBlock 34, 0.5) | Sup. / IN | Trained | 66.19% | HPARAM |
| ResNet-50 (DropBlock 34, 0.6) | Sup. / IN | Trained | 68.73% | HPARAM |
| ResNet-50 (DropBlock 34, 0.7) | Sup. / IN | Trained | 70.95% | HPARAM |
| ResNet-50 (DropBlock 34, 0.8) | Sup. / IN | Trained | 73.33% | HPARAM |
| ResNet-50 (DropBlock 1234, 0.1) | Sup. / IN | Trained | 27.19% | HPARAM |
| ResNet-50 (DropBlock 1234, 0.2) | Sup. / IN | Trained | 47.27% | HPARAM |
| ResNet-50 (DropBlock 1234, 0.3) | Sup. / IN | Trained | 55.59% | HPARAM |
| ResNet-50 (DropBlock 1234, 0.4) | Sup. / IN | Trained | 60.63% | HPARAM |
| ResNet-50 (DropBlock 1234, 0.5) | Sup. / IN | Trained | 64.57% | HPARAM |
| ResNet-50 (DropBlock 1234, 0.6) | Sup. / IN | Trained | 67.68% | HPARAM |
| ResNet-50 (DropBlock 1234, 0.7) | Sup. / IN | Trained | 70.35% | HPARAM |
| ResNet-50 (DropBlock 1234, 0.8) | Sup. / IN | Trained | 72.73% | HPARAM |
| ResNet-50 (Learning Rate 0.01) | Sup. / IN | Trained | 69.88% | HPARAM |
| ResNet-50 (Learning Rate 1.0) | Sup. / IN | Trained | 73.53% | HPARAM |
| ResNet-50 (Weight Decay 0.001) | Sup. / IN | Trained | 72.06% | HPARAM |
| ResNet-50 (LR 0.01, WD 0.001) | Sup. / IN | Trained | 73.90% | HPARAM |

Table 4: **Other Models**. These hyper-parameter sweeps were trained, but did not fit our target 74-78% accuracy range, so we did not use them in our main analyses. "Sup." indicates supervised learning, "IN" indicates ImageNet.

concatenated. In the second part, the two representations are first concatenated then dimensions are selected based on the diversity criteria.

## A.6 EXTRA FIGURES FOR REVIEWERS

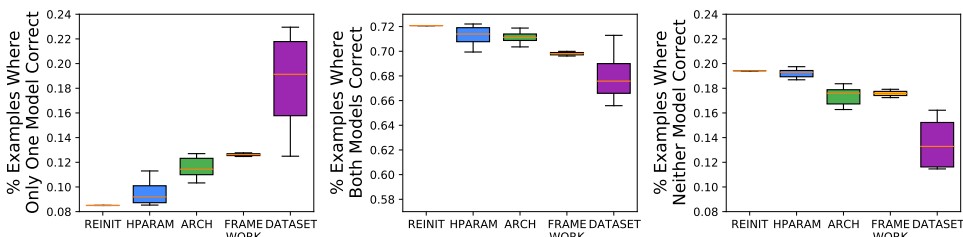

Figure 8: **As training methodology diverges, errors become uncorrelated**. The number of test examples where only one model predicts correctly increases (Left, same as "Error Inconsistency" in Fig. 1). Similarly, the number of examples where both models predict correctly (Center) and neither model predicts correctly (Right) decreases. For the most diverse training methodology ("Dataset", purple), we find more examples with inconsistent errors (left) than examples where both models predict correctly (right). This means that for an ensemble to perform better than another, it needs to resolve these inconsistent errors efficiently, to compensate for the decrease in the examples in the center plot.

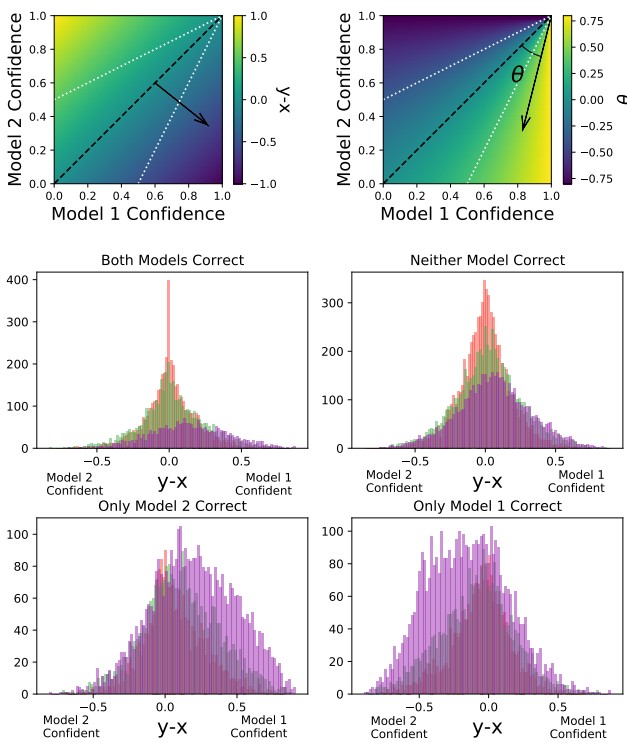

Figure 9: **Justification for** $\theta$. The white dotted lines represent the thresholds where an example's ensemble prediction will correspond to the higher-confidence-model's top-1 prediction. Theta measures distances in a way that aligns with these thresholds, allowing us to visualize the number of correctly-classified examples at a glance (by looking at the number of examples to the right/left of the dotted lines in Fig 3). Because y-x does not align with this threshold, it's harder to visualize the effect of specialization on performance, as can be seen by the histograms.

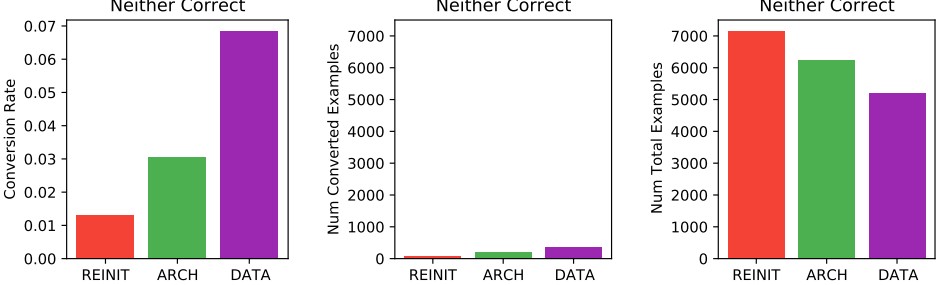

Figure 10: **Conversion rate of examples where "neither model" is correct individually**. We find that, as training methodology diverges, the conversion rate of these examples increases: Reinit (94 converted out of 7154 examples), Architecture (190 / 6246), Dataset (355 / 5192). This is consistent with our finding that different training methodologies create more efficient ensembles. We note that despite the higher conversion rate, a bigger effect on performance is in the decrease of examples where neither is correct (7154 -> 6246 -> 5192).

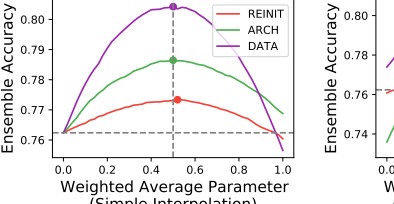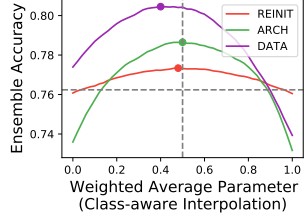

Figure 11: **Interpolating between ensemble members**. *Left*: We interpolate between the two models in the ensemble using a single parameter, and find that the optimal interpolating parameter seems to be at or near 0.5 (i.e.: it's just as good to simply average the logits with equal weight). We suspect this is because the calibration performed on the models before ensembling guarantees that the confidences are appropriately scaled already. *Right*: In order to further investigate whether specialization can yield better-weighted ensembles, we also perform a class-aware interpolation, where t=0 means nature classes get their logits from ResNet (and human classes get their logits from the other model), and vice versa for t=1. We find that for the ResNet+CLIP ensemble, t < 0.5 is optimal, which is consistent with our finding of ResNet/CLIP specialization. We note however that the boost is marginal (80.424% -> 80.452%, less than 15 images).

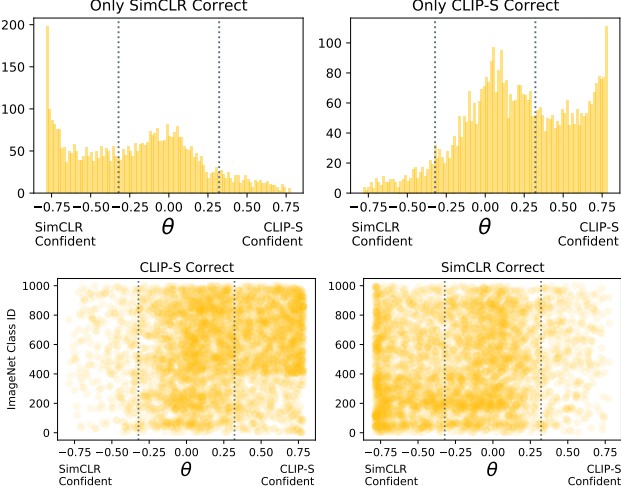

Figure 12: **Specialization of CLIP-S and SimCLR**. We find that these two models seem to also specialize in a way that is aligned with anthropogenic/nature classes (cids 500-900, and 0-300 respectively). We suspect the distribution of CLIP's pre-training dataset has a huge effect in making this happen. We also stress that these two models are still in very distinct categories: when comparing against each other (and not against our base model), we must note that they are different in reinit, architecture, params, and dataset, even if framework is kept the same.

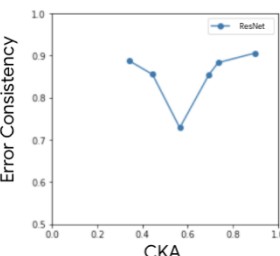

Figure 13: **CKA vs Error Consistency**. Here, CKA is measured between the two pre-logit representations of ResNet, and each of the models ResNet, CLIP-S-ZS, SimCLR, EfficientNet, ViT-B/16 and DenseNet-121. CKA and Error Consistency are not well correlated, likely due to CKA being sensitive to the geometry of representations.

