# OpenReview forum: "No One Representation to Rule Them All: Overlapping Features of Training Methods"
_ICLR.cc/2022/Conference — ICLR 2022 Poster_

### Official Review · Reviewer_7RNN · 2021-11-01

**Correctness:** 4
**Technical Novelty And Significance:** 3
**Empirical Novelty And Significance:** 3
**Recommendation:** 6
**Confidence:** 4

**Main Review:**

I find this paper well-motivated, with a clear focus on understanding whether different training methodologies learn similar representations, or not. The flow of the paper is well written, with each section addressing a one line of inquiry and naturally leading to the next. The (many) experiments are well designed, and the results clearly presented and interpreted. I especially enjoyed the first set of results (Figure 1), where the authors show that increasingly different training methodologies yield increasingly different representations — those results should be of interest to the ICLR community.

I list below some questions and concerns, in no particular order.

- If possible, please move Table 1 from the Appendix to the main text. I did not understand the experimental setup in Section 3.1 until I found this table.
- On p. 3, the paper introduces categories of possible changes to their base learning setup (supervised learning of Resnet-50 on ImageNet), and argues that some changes are more important than others: eg, changing the architecture to ViT is more important than doubling the weight-decay coefficient. Could the authors justify why changing the dataset is more important than changing the learning framework? It seems reasonable the procedure to infer model parameters (ie, the framework) can be as or more important than the choice of datasets, provided those datasets are large enough. If that's the case, we'd expect error inconsistency to be higher for "framework" than for "dataset" in Table 1, thus more diverse models are obtained by changing the framework (not the dataset). This begs the question: how much of the subsequent analysis depends on "dataset" being more important than "framework"? Would we see that better ensembles are built from models trained on the same (large enough) dataset but with different frameworks?
- The authors already provide experiments under many different conditions (see Table 1), and I do not wish to ask for more. Still, I question:
    - Why not include a Resnet-50 model when changing the framework and the dataset? This way, one could see how much improvement is expected when changing framework / dataset but the architecture is kept constant.
    - Are the (only) 2 different conditions enough to draw conclusions on the effect of changing the framework? One could imagine also including generative (auto-encoders, flows, etc) or few-shot training (protonet, metaoptnet, etc) as well.
- One of the main weaknesses of the paper is the lack of driving hypothesis. The experiments show that different training conditions yield different representations (and that those representations become specialized, yield better ensembles), but we're never told why. The second-to-last paragraph (p. 9) is not illuminating in that regard.
- Finally, there are a few moments of carelessness in the manuscript that need to be polished. For example, citations for WIT and JFT are never given; the "LBFGS Accuracy" of Section 4.6 is never properly defined (only mentioned in passing in Section 4.7); the section on bootstrapping (Section 4.5) doesn't discuss Freund and Schapire's work on weak learners; "Stochastic Weight Averaging" (Izmailov et al., UAI 2018) is never mentioned although it is a successful method to ensemble deep networks.

Some typos:

- p. 2: Use \citep for Andreassen et al. (2021)?
- p. 3: Is it 82 models since those in Table 4 are not included (looks more like 55-ish in Tables 1 and 2)?
- p. 4: "the the ensemble prediction".
- p. 8: "imagenet" not capitalized.
- p. 9: "CLIP-S is yields the most prediction diversity".
- p. 9: "any single models' accuracy" → "model's".
- p. 11: The reference is just "Lehman & Stanley", no "et al.".

**Summary Of The Paper:**

This paper proposes an empirical study of how different modelling choices (eg, hyper-parameters, architecture, training algorithm, dataset) result in different and complementary data representations. The first half of the paper shows that those choices influence model predictions, leading to models that specialize to a subset of the data. The second half shows that those different representations are complementary and can be ensembled to yield better performing models. In particular, the authors show that ensembling models that are most diverse gives the largest improvement when transferring to a downstream task.

**Summary Of The Review:**

Overall I remain positive about this paper and vote for acceptance. My main concerns are points 2 and 3 from the main review.

---

> ### Author Response · Authors · 2021-11-22
> **Reviewer response**
>
> We thank the reviewer for the positive review! We are excited that all reviewers see the paper favorably, and we appreciate the constructive feedback given. We respond to specific concerns below.
>
> > If possible, please move Table 1 from the Appendix to the main text
>
> Thank you for the feedback! It’s good to know that this table is useful to the reader in understanding our work! While it’s difficult to move it to the main text due to space constraints, we have updated the submission to feature the table’s existence more prominently.
>
> > Could the authors justify why changing the dataset is more important than changing the learning framework?
>
> Apologies for the confusion, we would like to clarify that we did not intend to make claims about the importance of each methodology change. In Sec 3.1 we talk about how some changes encompass others. For instance, changing the architecture incurs a necessary change to reinitialization parameters. Additionally, in order to guarantee optimal performance for this architecture, we also believe it is necessary to change regularization parameters (weight decay, etc). In that spirit, we note that the model categories we describe can be thought of as containing one another (and in this sense, each subsequent category represents increasingly-divergent training methodology).
> The reviewer makes a very good point that it is also possible to think of framework/dataset categories as orthogonal. Indeed, when we look at the ViT-S/16 model (trained supervised on ImageNet-21k and finetuned on ImageNet), we find a similar range of error inconsistency/ensemble boost as when we look at framework models (eg SimCLR). Still, our larger point that different training yields more uncorrelated errors can be seen by the fact that models trained with _both_ framework and dataset changes yield the most uncorrelated errors (and by extension ensemble performance).
> We realize that our current category structure allows for the misinterpretation that dataset is a “more important” category. Would splitting the “dataset” category into “dataset” and “dataset+framework” categories satisfy the reviewer’s concerns?
>
> > Why not include a Resnet-50 model when changing the framework and the dataset?
>
> SimCLR uses a ResNet backbone, so the results with SimCLR should actually show the effect of framework without changing the architecture.
>
> > Are the (only) 2 different conditions enough to draw conclusions on the effect of changing the framework?
>
> We agree with the reviewer that it would be really interesting to investigate the effect of other framework changes on prediction diversity. In our work, we’re attempting to demonstrate that there exist framework changes that have a bigger impact on model behavior than any architecture/hyper-parameter changes (categories for which we have extensive model variations), even if _not all_ framework changes will be good. By showing that none of the changes in earlier categories yield higher error inconsistency than a good framework change, we demonstrate our larger point that, when controlled for accuracy, methodology choices deeply affect model behavior.
>
> > The second-to-last paragraph (p. 9) is not illuminating in [motivating the work]
>
> We appreciate the feedback, and hope to clarify: one of our key motivations was to demonstrate how a single training methodology is not strictly better than all others (e.g.: no one methodology learns a ‘superset’ representation, not do they ‘dominate’ the predictions of lower accuracy models). Our results show that not only does methodology choices deeply affect model behavior, but that the best performance combines objectives that don’t directly optimize ‘accuracy’.
> The implications of this are that we shouldn’t assume SGD will optimally find the best solution (even when the loss directly maximizes the metric we care about), nor  should we assume the highest accuracy model is always the best solution. Finally, our results show the field should prioritize finding more creative solutions to the tasks we care about (and deeply understanding those tasks), rather than continuously exploiting known setups with minor changes (e.g.: architecture changes) that may only yield marginal accuracy improvements.
>
> > Polishing/Typos
>
> We thank the reviewer for pointing these out, we have fixed them in the submission.
>
> Thank you once again for your time reviewing our paper, and for the useful feedback. We feel like your feedback has helped us improve our paper, and hope to have satisfied your concerns.

---

> > ### Comment · Reviewer_7RNN · 2021-11-28
> > **Thank you for the replies.**
> >
> > Thank you for the replies. I keep my positive score and hope that some clarifications can be included in the final version of the paper. Below are some additional comments based on the response.
> >
> > * Re: "Would splitting the “dataset” category into “dataset” and “dataset+framework” categories satisfy the reviewer’s concerns?"
> >     * Thanks for the discussion, which I largely agree with. I don't think such splitting is necessary, provided your clarification is clearly conveyed in the main text. In particular, I believe it would strengthen the paper's claim if you (quantitatively) underlined that changing both framework and dataset outperforms changing only one of them.
> >     * The paper prompts one additional question, namely: how to choose over two diversification axes (eg, dataset vs framework) if they are orthogonal to each other? I believe this can be left to future work.
> > * Re: "only 2 different training frameworks"
> >     * I remain somewhat unconvinced by the explanation for including only 2 training frameworks, especially since you state in the next paragraph that your motivation "was to demonstrate how a single training methodology is not strictly better than all others". In that case, including more framework variations (and showing they result in better ensembles than any single framework) would strengthen your argument. While you've done it with datasets, I'd argue a similar result for frameworks would be more surprising.

---

### Official Review · Reviewer_yy7C · 2021-11-02

**Correctness:** 4
**Technical Novelty And Significance:** 2
**Empirical Novelty And Significance:** 4
**Recommendation:** 8
**Confidence:** 4

**Main Review:**

This paper is well-written and interesting, and the empirical analysis is impressive. The authors present their empirical observations as the main contribution, but I think this actually undersells the contributions -- this paper is also a tour de force in methodologies about how to evaluate and compare different models. Section 4 is very well-written and is a golden example of strong experimental design/scientific method; each sub-section presents a concrete hypothesis, an experiment to test the hypothesis, and discussion of empirical results. I wish every paper were written this well!

My only complaint is that some of the language is imprecise. As a concrete example, the phrase "different enough" or "diverse enough" appears several times throughout the manuscript. This phrase can be more precise (What does it mean to be "different enough?" Different architecture? Different dataset?). The results speak for themselves by the end of section 4, but the takeaways could be more precise for the reader (note that this may take more analysis/experiments -- but I'm sure the authors can knock that out of the park).

I also think the paper could be improved by including some more intuition or examples of the types of divergences between different models. For example, what types of examples does CLIP get wrong but SimCLR gets right? This may be unwieldy for all 82 models, but a few spot instances for each framework would be very interesting and useful.

**Summary Of The Paper:**

This paper presents an extensive empirical evaluation of what models learn when trained with different architectures, frameworks, and datasets. The authors discuss the effects of training methodology on the types of errors that models make, and show that ensembling models trained with diverse training methodologies can result in surprisingly strong performance (even when the models in the ensemble themselves are relatively weak).

**Summary Of The Review:**

This paper is interesting and well-written, and it presents an impressive empirical evaluation of how training methodology affects representations and predictions. The paper does not claim novel methods, but the methods for empirical analysis themselves are interesting, and the community can learn a lot from them.

---

> ### Author Response · Authors · 2021-11-22
> **Reviewer response**
>
> We thank the reviewer for the positive review! We are excited that all reviewers see the paper favorably, and we appreciate the constructive feedback given. We respond to specific concerns below.
>
> > My only complaint is that some of the language is imprecise. … What does it mean to be "different enough?"
>
> Thank you for pointing this out, and apologies for the confusion. In Sec 3.1 we talk about how some changes encompass others. For instance, changing the architecture incurs a necessary change to reinitialization parameters. Additionally, in order to guarantee optimal performance for this architecture, we also believe it is necessary to change regularization parameters (weight decay, etc). In that spirit, we note that the model categories we describe can be thought of as containing one another (and in this sense, each subsequent category represents increasingly-divergent training methodology). By “different enough” we mean that the number of methodology changes made is in a category that incurs more of these changes.
>
> More specifically, in Fig 5, for a standard-accuracy model to be useful when ensembling with a high-accuracy model, it needs to have at least some amount of error inconsistency (with respect to the high accuracy model). To obtain such high error inconsistency, the best ensemble members’ methodologies differ the most (changing architecture, framework, _and_ dataset). We have updated the paper to better discuss this ordering of categories, and hope this helps clarify the language as well.
>
> > For example, what types of examples does CLIP get wrong but SimCLR gets right?
>
> We thank the reviewer for the incredible suggestion! We run this specialization analysis for SimCLR and CLIP, and add it to the Appendix Fig 12. We find that these two models seem to also specialize in a way that is aligned with anthropogenic/nature classes. We suspect the distribution of CLIP’s pre-training dataset has a huge effect in making this happen. We also stress that these two models are still in very distinct categories: when comparing against each other (and not against our base model), we must note that they are different in reinit, architecture, params, and dataset, even if framework is kept the same. We think running similar analysis for more models is an interesting direction, that comes with its own challenges: for instance, in the case of ResNet + EfficientNet, the specialization still occurs (Fig 3) but it does not fall along class ID lines (see Fig 4). Figuring out exactly what types of examples each model specializes in could be its own exploratory inquiry.
>
> Thank you once again for your time reviewing our paper, and for the useful feedback. We feel like your feedback has helped us improve our paper, and hope to have satisfied your concerns.

---

> > ### Comment · Reviewer_yy7C · 2021-11-29
> > **Rebuttal Response**
> >
> > Thank you for your detailed response and for incorporating some suggested changes in the paper. I still believe that the paper is accepted, so I will be keeping my score.
> >
> > I think "different enough" still appears as a phrase in the current version of the paper that I see -- for the camera ready, I think the language could be a bit stronger by removing this phrase (and just listing exact differences that you see lead to benefits). I'm not completely sure how to read figure 12 in Appendix A. It seems like an interesting data point, but I'm not sure how to read "anthropogenic/nature classes" from the figure axes -- may be worth adding some more details in the captions/axes to help readers!

---

> > > ### Author Response · Authors · 2021-11-30
> > > **Response**
> > >
> > > Thank you for the feedback! We will update that language, and the details in the Appendix figure for the camera-ready version :)

---

### Official Review · Reviewer_ZeBH · 2021-11-02

**Correctness:** 3
**Technical Novelty And Significance:** 4
**Empirical Novelty And Significance:** 4
**Recommendation:** 8
**Confidence:** 4

**Main Review:**

I find this paper very insightful and believe that the investigations done here would be useful for the community.  Ensembling is a prevalent way of achieving desired accuracies in practice and this study provides helpful pointers to do that effectively. I have following questions for the authors though:

- It might just be me, but how is the training methodology "Frameworks" is incorporated in this work? I could not find any details or explanation about it.
- In the frameworks section, SimCLR is used. This model is trained with a self-supervised objective with no prediction head for labels. How do you attain prediction confidences on a test set for such models? or how do you incorporate predictions for such a model?
- Based on the previous question, is it possible to use different self-supervised or even unsupervised methods?
- The dataset difference is a part of the study, but is it fair to compare the accuracies on image-net for the models which were not trained on image-net? Wouldn't the methods which aren't trained on image-net suffer from domain generalization problem?
- In CLIP-S method, it is unclear to me whether it's the training dataset that introduces a huge divergence in the methodology or the contrastive objective. I think there could be a sub-categorization of the training methodology "Dataset" into "optimization objectives" and "datasets" for a more controlled study. I will expand on this in the following point.
- Figure 4, demonstrates some clear differences in the specialization of CLIP-S and resnet-50. Since the models are trained on different datasets and then tested on image-net. I wonder how much important the bias of the training dataset is. More importantly, what will happen if the ResNet is trained on the [somewhat similar] dataset used for training CLIP-S and tested on image-net? would the variation in architecture and frameworks still matter?
- In the light of the above, supervised contrastive learning [1] might be a useful model to try as it uses contrastive training objective and is trained on image-net.
- Have you quantified the conversion rate where both ensembling models predict incorrectly. I wonder if the ensembled prediction confidences might make the incorrect prediction correct. It will be interesting to see how various training methodologies influence that?
- Is there an opportunity for weighted ensembling? Based on the analysis of training methodology can one further improve the ensembling by rescaling prediction confidences of certain models for certain test datasets? One can probably use specialization criteria to do that?
- In section 4.6, do you train a linear classifier (via L-BFGS) on top of concatenated representations? How do you pick the portions of each representation to be concatenated? is it selected randomly?
- In section 4.6 (later half), do you select diverse features from both models at the same time and concatenate these features or you do that separately for both models?

[1] [https://arxiv.org/abs/2004.11362](https://arxiv.org/abs/2004.11362)

**Summary Of The Paper:**

This paper empirically investigates how different models, trained with different methodologies should be ensembled to maximize the accuracy in image-net classification task. By carefully designed experiments, the authors look at different aspects of the trained models and provide guidelines for selecting models to be ensembled. The main takeaway: models trained with increased divergence in training methodologies are best suited for ensembling.

**Summary Of The Review:**

I liked the overall study and believe that it will be useful for the community. I would appreciate it if the authors could address my concerns/ questions above.

---

> ### Author Response · Authors · 2021-11-22
> **Reviewer response**
>
> We thank the reviewer for the positive review! We are excited that all reviewers see the paper favorably, and we appreciate the constructive feedback given. We respond to specific concerns below.
>
> > how is the training methodology "Frameworks" is incorporated in this work?
> How do you attain prediction confidences on a test set for such models?
>
> The models in the Frameworks category differ from the base model in that they are (pre)trained with non-supervised losses. In particular, SimCLR, CLIP, and ALIGN are trained with a contrastive loss. One way to think about it is that models in the Reinit, Hyper-parameter, and Architecture categories are never pre-trained on different losses/datasets, whereas Framework and Dataset models are allowed to be pre-trained. Once they are pre-trained, we obtain imagenet classification logits via a variety of methods (depending on the model). For instance, SimCLR is finetuned on ImageNet, whereas CLIP-L-ZS has a zero-shot classification head that yields imagenet-class predictions without the need of finetuning. In all cases, we make sure that the logits obtained yield accuracy that lies within our target accuracy range of interest, in order to guarantee fair comparison.
>
> > Wouldn’t the [dataset-category models] which aren't trained on image-net suffer from domain generalization problem?
>
> Throughout our analysis, we make sure that all models used are within a narrow accuracy range, which allows us to make sure the effects we’re observing are indeed a result of methodology change (and not of a single model being particularly higher accuracy than others, for instance). This means that the models in the dataset category, even when not finetuned on ImageNet (e.g.: CLIP-L-ZS, which uses a zero-shot prediction head), still have comparable accuracy to the other models. Indeed seem to be the models with the most inconsistent errors, and therefore the models that provide the highest benefit in ensembling, further indicating that they’re not suffering from domain generalization.
> This is perhaps because the other datasets used (WIT, JFT) are large, and relatively similar to imagenet (i.e.: they are not of a narrower distribution, like a medical imaging dataset would be). We agree with the reviewer that further studying the effect of pre-training distribution would be an interesting line of inquiry, but we believe it is beyond the scope of our work.
>
> > unclear to me whether it's the training dataset that introduces a huge divergence in the methodology or the contrastive objective
>
> We agree with the reviewer determining the exact effect of frameworks vs datasets is an interesting question to explore. The reviewer makes a very good point that it is also possible to think of framework/dataset categories as orthogonal. Indeed, when we look at the ViT-S/16 model (trained supervised on ImageNet-21k and finetuned on ImageNet), we find a similar range of error inconsistency/ensemble boost as when we look at framework models (eg SimCLR). Still, our larger point that different training yields more uncorrelated errors can be seen by the fact that models trained with _both_ framework and dataset changes yield the most uncorrelated errors (and by extension ensemble performance).
> We realize that our current category structure allows for the misinterpretation that dataset is a “more important” category. We’d be happy to split the “dataset” category into “dataset” and “dataset+framework” categories satisfy the reviewer’s concerns.
> We thank the reviewer for the suggestion to test the “supervised contrastive learning” model. We will contact the authors to obtain logits and incorporate this work into our paper.
>
> > Have you quantified the conversion rate where both ensembling models predict incorrectly
>
> Thank you for the suggestion! We have included this analysis in the Appendix. We find that indeed differently-trained ensembles exhibit higher conversion rates in examples where neither model predicts correctly: Reinit (94 converted out of 7154 examples), Architecture (190 / 6246), Dataset (355 / 5192). We note that despite the higher conversion rate, a bigger effect on performance is in the decrease of examples where neither is correct (7154 -> 6246 -> 5192). See Fig 10 for details.
>
> (continued)

---

> > ### Author Response · Authors · 2021-11-22
> > **Reviewer response (part 2)**
> >
> > (cont.)
> >
> > > Is there an opportunity for weighted ensembling?
> >
> > Thank you for the insightful suggestion! We ran a few preliminary analyses to get at this question, and have added them to the Appendix Fig 11.
> > In particular, we tried interpolating between the two models in the ensemble using a single parameter. We find that the optimal interpolating parameter seems to be at or near 0.5 (i.e.: it’s just as good to simply average the logits with equal weight). We suspect this is because the calibration performed on the models before ensembling guarantees that the confidences are appropriately scaled already.
> > In order to further investigate whether specialization can yield better-weighted ensembles, we also perform a class-aware interpolation, where t=0 means nature classes get their logits from resnet (and human classes get their logits from the other model), and vice versa for t=1. We find that for the Resnet+CLIP ensemble, a t < 0.5 is optimal, which makes sense given our finding of ResNet/CLIP specialization. We note however that the boost is marginal (80.42399088541666% -> 80.45247395833334%, less than 15 images).
> >
> > > [how] do you train a linear classifier (via L-BFGS) on top of concatenated representations?
> >
> > In order to train the linear classifiers, we use L-BFGS in the same setup/implementation described in Kornblith et al. (2019). We train the linear heads on top of the pre-logit representations without augmentation. We find it useful to normalize each representation before any operations (concatenation, subsampling, etc). In the first part of Section 4.6, the portions of each representation are picked randomly (e.g.: random 25% dimensions of ResNet, and random 75% dimensions of CLIP), and then concatenated. In the second part, the two representations are first concatenated then dimensions are selected based on the diversity criteria. We have added this information to the paper for completeness.
> >
> > Thank you once again for your time reviewing our paper, and for the useful feedback. We feel like your feedback has helped us improve our paper, and hope to have satisfied your concerns.
> >
> > Kornblith, Simon, Jonathon Shlens, and Quoc V. Le. "Do better imagenet models transfer better?." Proceedings of the IEEE/CVF Conference on Computer Vision and Pattern Recognition. 2019.

---

> > > ### Comment · Reviewer_ZeBH · 2021-11-28
> > > **Post Rebuttal**
> > >
> > > I thank the authors for the detailed response. I also appreciate the additional analysis and believe they add more value to the work.
> > >
> > > _Regarding splitting the “dataset” category into “dataset” and “dataset+framework” categories_
> > >
> > > Though it's not a deal-breaker, I think splitting it would be a good idea. I would help the practitioners in making a decision about the datasets when doing pretraining.
> > >
> > > I believe the work is useful for the community and thus recommend its acceptance. This is also reflected in my updated score.

---

### Official Review · Reviewer_iemd · 2021-11-04

**Correctness:** 3
**Technical Novelty And Significance:** 4
**Empirical Novelty And Significance:** 4
**Recommendation:** 8
**Confidence:** 3

**Main Review:**

Strengths:

The paper is well-written, and has clearly very thorough experiments. Notably, they include 82 different models (across their 5 categories of training diversity), which include recent high-performing works.

Each point in their 5 listed contributions at the end of the introduction is interesting and valuable for future research (although I have some concerns listed below). Each point provides evidence for the authors' concluding statement, that no single training methodology should dominate the field, but rather research effort on a diverse set of approaches should continue. They also raise a very interesting question for theory as well as empirical studies - why doesn't SGD learn the more diverse representations that are shown to perform better?

One part of "well-written" - the authors repeat key definitions throughout the work, instead of defining them once, which I found helpful (e.g. repeating the definition of "conversion rate" in the Figure 2 caption).

Weaknesses:

Uncorrelated errors seem like an intuitive metric to analyze representation diversity - however, there are other existing metrics which I imagine measure this more directly, such as mutual information. Why not measure representation diversity in that manner?

I am also not sure why the authors chose to use the angle distance on the confidence-confidence plot in their analysis on model specialization. It works, but seems a little odd - why not just subtract one model confidence from another?

In Table 3 in the Appendix, the authors seem to define "low accuracy" models as those below the 74% accuracy (on ImageNet) threshold they use in their main analysis. However, Section 4.5 considers the 74%-78% range as "low accuracy" - which is it? In the current form, the claim "low accuracy models can benefit high-accuracy ensembles" is misleading.

A small clarification question - why is the dataset star in Figure 2 marked with an additional cross? To highlight the best category? If so, that would be useful to mention (assuming I didn't miss the explanation somewhere).

**Summary Of The Paper:**

The authors conduct a large-scale study on when models are more likely to learn different representations, and how diversity in learned representations can help ensemble performance. They select 5 possible reasons for representation diversity: model reinitialization, hyperparameters, model architectures, model frameworks, and datasets. They define a measure of error correlation, and show that a pair of models with more uncorrelated errors results in higher ensemble performance - even if one or both of those models has "low accuracy". They show that models trained on different datasets are most likely to have uncorrelated errors, as opposed to the other sources of representation diversity. The authors study the effect of sampling and concatenating features from two different models, find that a mix of features from both models yields the best performance, and conclude that the models have learnt different features (yet likely overlapping). They also examine the categorical specializations between varying models, as well as the effect of training diversity for creating ensembles on downstream tasks.

**Summary Of The Review:**

The authors present a thorough evaluation into the effect of various differences in training procedures on resulting representation diversity. I have not specialized in this research area and cannot speak well to the novelty of each claim, but believe that the paper deserves acceptance due to the extensive experiments, clear presentation, and interesting findings which open questions for future research.

---

> ### Author Response · Authors · 2021-11-22
> **Review response**
>
> We thank the reviewer for the positive review! We are excited that all reviewers see the paper favorably, and we appreciate the constructive feedback given. We respond to specific concerns below.
>
> > Why not measure representation diversity [with mutual information]?
>
> We thank the reviewer for the suggestion! We agree that finding measures in representation space to quantify the diversity is an interesting question. However, measuring mutual information in representation space is a challenging problem (McAllester & Stratos 2018) and has led to erroneous results in prior work (Saxe et al., 2018).  As an alternative, we have included Fig 13 in the Appendix showing the correlation of CKA (Kornblith et al., 2019) of representations with our measure of error consistency. We find that CKA is not well correlated with error consistency, likely due to CKA being sensitive to the geometry of representations.
>
> We opt to use error inconsistency because, not only is it an intuitive measure, but also it allows us directly connect the representation diversity with their performance benefit, since error inconsistency directly quantifies the portion of the validation set where prediction inconsistency needs to be resolved by the ensemble procedure.
>
> > [why] use the angle distance? why not just subtract one model confidence from another?
>
> We appreciate the suggestion! The two measures are geometrically related: y-x is the distance between the point and the y=x line, whereas theta measures the angular distance between the point and the y=x line. We have added Fig 9 in the Appendix to visualize the differences between them. Notably: in the confidence-confidence plot, the white dotted lines represent the thresholds for the regions where an example’s ensemble prediction will correspond to the higher-confidence-model’s top1 prediction. Theta measures distances in a way that aligns with these thresholds. This means that in the histograms in Fig 3 it is possible to visualize the number of examples correctly classified at a glance (by looking at the number of examples to the right/left of the dotted lines). Because y-x does not align with this threshold, it’s harder to visualize the effect of specialization on performance, as can be seen by the histograms using y-x in Fig 9.
>
> > "low/high” accuracy models
>
> We thank the reviewer for pointing this out. The 74-78 accuracy range is the “standard accuracy” one expects from a typical imagenet model. That's the range we use for most of our analysis. Models with higher accuracy than this are usually among the best-performing imagenet models available – we term these “high accuracy”. Practitioners often look to these high accuracy models as starting points for finetuning on downstream tasks. So in a relative sense, standard accuracy models are often times seen as "low accuracy". What we show is that they indeed are useful. We additionally show that very-low accuracy models (which we term “low accuracy” due to their lower accuracy than the standard models) can also be useful. We apologize for the confusion. We clarified this discussion in the paper to be consistent with the “low/standard/high” accuracy verbiage.
>
> > why is the dataset star in Figure 2 marked with an additional cross?
>
> Apologies for the confusion. The cross marks the standard error of the category, since we average over the many models within a category. We have clarified this in the caption.
>
> Thank you once again for your time reviewing our paper, and for the useful feedback. We feel like your feedback has helped us improve our paper, and hope to have satisfied your concerns.
>
>
> Kornblith, Simon, et al. "Similarity of neural network representations revisited." International Conference on Machine Learning. PMLR, 2019.
> McAllester, David, and Karl Stratos. "Formal limitations on the measurement of mutual information." International Conference on Artificial Intelligence and Statistics. PMLR, 2020.
> Saxe, Andrew M., et al. "On the information bottleneck theory of deep learning." Journal of Statistical Mechanics: Theory and Experiment 2019.12 (2019): 124020.

---

### Decision · Program_Chairs · 2022-01-20

**Decision:**

Accept (Poster)

**Comment:**

This well written and well motivated paper has been independently reviewed by four expert reviewers. They all voted for the acceptance with three straight accepts and one marginal. The feedback provided to authors was constructive and the authors responded comprehensively. I recommend acceptance of this work for ICLR.